# Inflammation of Conduction Tissue in Patients with Arrhythmic Phenotype of Myocarditis

**DOI:** 10.3390/jcm9113470

**Published:** 2020-10-29

**Authors:** Andrea Frustaci, Romina Verardo, Maria Alfarano, Cristina Chimenti

**Affiliations:** 1Department of Clinical, Internal, Anesthesiologist and Cardiovascular Sciences, Sapienza University, 00161 Rome, Italy; maria.alfarano@uniroma1.it (M.A.); cristina.chimenti@uniroma1.it (C.C.); 2Cellular and Molecular Cardiology Lab, IRCCS L. Spallanzani, 00149 Rome, Italy; romina.verardo@inmi.it

**Keywords:** arrhythmic phenotype, conduction tissue, myocarditis

## Abstract

Background: Myocarditis can manifest with lone ventricular tachyarrhythmias (LVT). Elective inflammation of conduction tissue (CT) is supposed but unproved. Methods: Forty-two of 420 patients with biopsy proven myocarditis presented with LVT. Twelve of them included CT sections in endomyocardial biopsies. Real-time polymerase chain reaction (PCR) for viral genomes, immunohistochemistry for viral antigens and Toll like receptor 4 (TLR4) were performed. Twelve myocarditis patients with infarct-like or cardiomyopathic phenotype and CT included in tissue section were used as controls. Results: Four of the 12 patients presented non-sustained ventricular tachycardia (nsVT), six with sustained ventricular tachycardia (sVT), two with ventricular fibrillation. CT was inflamed in all LVT patients and not in controls (*p <* 0.001). PCR was positive for influenza-A virus in two, herpes simplex virus type 2 (HSV2) in one and adenovirus in one with positive CT immunostaining for viral antigens. In eight patients, negative PCR and TLR4 overexpression suggested an immune-mediated pathway. Patients with influenza-A myocarditis and CT infection responded to oseltamivir, those with HSV2 (Herpes Virus 2) and adenovirus infection died. The eight patients with immune-mediated myocarditis were treated with steroids and azathioprine. Seven of them had no more VT(ventricular tachyarrhythmias)during six-month follow-up. Conclusions: Arrhythmic phenotype of myocarditis is associated with CT inflammation/infection. Molecular characterization of CT damage may lead to pharmacologic control of arrhythmias in 75% of cases.

## 1. Introduction

Myocarditis can manifest with a variety of symptoms that can be incorporated into three major phenotypes: infarct-like, cardiomyopathic and arrhythmic type [1]. The first pattern is characterized by fever, chest pain, deflection of ST segment at the electro-cardiogram (ECG) and serologic increase in Troponin I that mimic an acute ischemic event. It is easily recognizable at cardiac magnetic resonance (CMR) because of common subepicardial/intramyocardial edema with abnormal increase of late gadolinium enhancement (LGE) and T2 signals. If cardiac function is preserved or recovers in a short time with supportive therapy, it doesn’t require further invasive investigations beyond coronary angiography.

Differently, cardiomyopathic phenotype has a more subtle course with progressive dilatation and dysfunction that looks like a dilated cardiomyopathy and where CMR sensitivity based on Lake Louise criteria [2] is reduced to 57% [3]. In this instance an endomyocardial biopsy is recommended to confirm the diagnosis and establish the most appropriate (anti-viral or immunosuppressive) therapy.

Finally, identification of myocarditis in patients presenting with lone ventricular tachyarrhythmias is the most difficult to be obtained as cardiac contractility is usually preserved and CMR sensitivity, even for technical reasons linked to electrical instability, is as low as 40% [3]. In this setting, the diagnosis remains often presumptive and the management is usually conservative based on antiarrhythmic drugs and/or implantable cardiovertebre defibrillator (ICD) implantation. However, both solutions present serious limitations as the former can be ineffective, exposing the patient to the risk of cardiac arrest while the latter can be excessive since a spontaneous resolution leaves the patient with a permanent unnecessary device. These limitations can be solved by a ventricular endomyocardial biopsy which may include sections of conduction tissue (CT) [4] and, through a morpho-molecular tissue characterization, provide indication to a specific therapy.

## 2. Patient Population

Among 420 patients with a biopsy proven myocarditis diagnosed from 2009 to 2019 in our institution, 84 (20%) included in left ventricular endomyocardial biopsies of sections of conduction tissue (CT). Forty-two presented with ventricular tachyarrhythmias and normal cardiac anatomy and function left ventricular ejection fraction (LVEF ≥ 50%), i.e., lone ventricular tachyarrhythmias (LVT). Of these, 12 patients presented with CT sections in endomyocardial biopsies. This cohort represents our study population. The control group was represented by 12 consecutive patients with myocarditis, infarct-like or cardiomyopathic phenotype and no repetitive ventricular tachyarrhythmias (Lown class ≤ 3) at Holter monitoring, matched for sex and age with LVT patients and presenting CT sections included in the endomyocardial tissue.

## 3. Material and Methods

### 3.1. Clinical Studies

All 12 patients underwent extensive clinical examination with non-invasive (resting ECG, Holter monitoring, 2D-echocardiogram and cardiac magnetic resonance (CMR), and invasive cardiac studies (cardiac catheterization, selective coronary angiography, left ventricular (LV) angiography, and LV endomyocardial biopsy).

Echocardiographic parameters were determined according to established criteria [5]. In particular, left ventricular ejection fraction (EF)was calculated in the apical 4- and 2-chamber views from three separate cardiac cycles using the modified Simpson’s method.

CMR was performed on a 1.5 Tesla scanner (Magnetom Avanto, Siemens Healthcare, Germany). Standard cardiac magnetic resonance protocol included: (i) cine magnetic resonance images acquired during breath-holds in the short-axis, 2-chamber, and 4-chamber; (ii) black blood T2-weighted short-tau inversion recovery images on short-axis planes covering the entire left ventricle during 6 to 8 consecutive breath-holds for myocardial edema detection; (iii) late gadolinium-enhanced imaging performed 15 min after injection of 0.2 nmol/kg of gadoterate meglumine and signal intensity value 2 SDs above the mean signal intensity of the remote normal myocardium was considered suggestive for myocardial fibrosis. Functional and morphological data were analyzed according to the Lake Louise criteria [2] for which CMR diagnosis of myocarditis is suggested by the presence of at least two of the following criteria, including early enhancement (hyperaemia as an equivalent of vasodilation), edema on T2-weighted/Short Tau Inversion Recovery (STIR) sequences, and late gadolinium enhancement (LGE) in a nonischemic pattern.

### 3.2. Endomyocardial Biopsy Studies

Cardiac catheterization with left ventricular and coronary angiography was obtained in all patients. Endomyocardial biopsy (four to eight samples for each patient) was performed in the septal-apical region of the left ventricle [6]. All subjects gave their informed consent for inclusion before they participated in the study. The study was conducted in accordance with the Declaration of Helsinki, and the protocol was approved by the Local Ethics Committee with number 980bis/17 on 30 October 2017 (Project identification code: FARM12JCXN, EudraCT number 2016-003014-28).

### 3.3. Histology, Electron Microscopy and Immunohistochemistry

For histological analysis, the endomyocardial samples were fixed in 10% buffered formalin and paraffin embedded. Five micron-thick sections were stained with hematoxylin and eosin and Masson thrichrome. For electron microscopy, additional samples were fixed in 2% glutaraldehyde in a 0.1 M phosphate buffer, at pH 7.3, post-fixed in osmium tetroxide and processed following a standard schedule for embedding in Epon resin. Ultrathin sections were stained with uranyl acetate and lead hydroxyde.

Histologic diagnosis of myocarditis included evidence of leukocyte infiltrates in association with damage of the adjacent myocytes, according to the Dallas criteria [7] confirmed by immunohistochemistry [1]. In particular, for the phenotypic characterization of the inflammatory infiltrates, immunohistochemistry for CD3, CD20, CD43, CD45RO and CD68 was performed (all Dako, Carpinteria, CA, USA). The presence of an inflammatory infiltrate ≥ 14 leucocytes/mm^2^ including up to 4 monocytes/mm^2^, with the presence of CD3-positive T-lymphocytes ≥ 7 cells/mm^2^ associated with evidence of degeneration and/or necrosis of the adjacent cardiomyocytes, was considered diagnostic for myocarditis.

For the assessment of Toll like receptor 4 (TLR4) expression, myocardial sections were incubated with goat antirabbit TLR4 antibody (Santa Cruz, CA, USA, 1:10), and a peroxidase/anti-peroxidase complex, followed by labeling with chromogen diaminobenzidine (Dako, Carpinteria, CA, USA). A semiquantitative evaluation of the immunoreactivity for TLR4 (grading from 0 to 4) was applied [8].

CT was identified at histology as loosely-arranged small myocytes, positive to HCN4 (Hyperpolarization-activated and cyclic nucleotide-gated) immunostaining [4] supplied by a centrally-placed thickened wall arteriole, circumscribed by a fibrous membrane in a fascicle configuration (Monckeberg [9] and Aschoff [10] criteria). HCN4 immunohistochemistry was performed on formalin-/paraformaldehyde-fixed paraffin-embedded sections of endomyocardial samples to identify HCN4 positive cells, as described [11]. Briefly, tissue sections were incubated with a rat monoclonal antibody at dilution of 1:20 recognizing HCN4 (Pierce Antibody Products, Thermo Fisher Scientific Inc, 3747 N. Merdian Road, Rockord, IL, USA) or without primary antibody (negative control) overnight at 4 °C in a humidified chamber. Detection was performed using a biotin-conjugated secondary antibody and streptavidin-horseradish peroxidase conjugate for enzyme immunoassay (SA-HRP) (UCS Diagnostics Srl., Morlupo, Italy) at room temperature for 10 min. Samples were washed and incubated by colorimetric detection using 3.3′-diaminobenzidine (DAB, Dako), counterstained with hematoxylin, and mounted with a quick-hardening mounting medium (Eukitt, Bio-OpticaSpA, Milano, Italy).

Immunohistochemistry for viral antigens was performed on formalin-/paraformaldehyde-fixed paraffin-embedded sections of endomyocardial samples.

### 3.4. Molecular Biology Studies

In all patients at baseline a Real time polymerase chain reaction (PCR) [12] for the most common cardiotropic viruses (adenovirus, enterovirus, influenza A and B viruses, Epstein Barr virus, parvovirus B19, hepatitis C virus, cytomegalovirus, human herpesvirus 6, herpes simplex virus types 1 and 2) was performed to determine whether a viral infection was the cause of myocarditis.

### 3.5. Statistical Analysis

Data are presented as counts for categorical variables and as mean ± standard deviation for numeric variables. Comparison among groups (LVT vs. controls) was performed using the Mann-Whitney U-test for numeric and the χ^2^ test (or Fisher where applicable) for categorical variables. A value of *p* < 0.05 was considered as significant. Correlation between CD3 positive cells in CT and Lown class in all patients (12 LVT and 12 controls) was explored with Spearman’s correlation. Data analysis was performed using GraphPad Prism version 6.04 for Windows (GraphPad Software, La Jolla, CA, USA).

## 4. Results

### 4.1. Clinical Studies

Clinical baseline data are summarized in Table 1.

The cohort mean age was 46.08 ± 14.81 and 58% (7/12) were males. Sixty-seven percent of patients had a flu-like syndrome before the onset of myocarditis.

2D-echocardiography showed normal cardiac parameters with preserved bi-ventricular function in all patients. CMR was performed in 11 of 12 patients, because the patient with herpes simplex virus type 2 (HSV2)-related myocarditis promptly died of ventricular fibrillation. Cardiac magnetic resonance images showed no signs of acute myocardial damage (absence of myocardial edema on T2-weighted images and normal native T1 values). A late gadolinium enhancement was detected in 54% (6/11) of patients, suggesting a mild interstitial fibrosis.

All patients had ventricular arrhythmias. Four patients presented with non-sustained ventricular tachycardia (nsVT), six with sustained ventricular tachycardia (sVT), one patient with arrhythmic storm and one patient died because of ventricular fibrillation.

### 4.2. Endomyocardial Biopsy Studies

To investigate the cause of electrical instability, all patients underwent an invasive cardiac study including coronary angiography and left ventricular endomyocardial biopsy. No complications resulted from the endomyocardial biopsy. Coronary network was unaffected in all patients. At histology, a focal myocarditis with scattered lymphomononuclear infiltrates (CD43+, CD3+, CD45RO+) associated with necrosis of the adjacent myocytes was observed. Biopsy samples also included sections of peripheral branches of conduction tissue with inflammatory infiltration and cell necrosis in all patients. CT was identified at histology as loosely arranged small myocytes, positive to HCN4 immunostaining (Figure 1G insert i) [4] supplied by a centrally-placed thickened wall arteriole, circumscribed by a fibrous membrane in a fascicle configuration (Monckeberg [9] and Aschoff [10] criteria) (Figure 2E). Real time PCR on frozen samples revealed a positivity for influenza A virus in two patients; adenovirus in one patient and herpes simplex virus type 2 (HSV2) in one patient. Immunostaining for the corresponding viral antigens (envelope glycoprotein B for HSV2, Figure 2F) showed positivity in CT sections. In the remaining 67% of patients (*n* = 8), negative PCR for viral genomes and TLR4 myocardial overexpression suggested an immune-mediated pathway.

In the control group represented by patients with myocarditis and no repetitive ventricular tachyarrhythmias, CT appeared devoid of inflammatory infiltration and damage (Figure 3).

### 4.3. Correlation

The intensity of CT inflammation (CD3+/mm^2^ T lymphocytes) correlated with the severity of arrhythmic manifestations (Lown class) (Spearman rho = 0.7791, *p <* 0.001) (Figure 4A). The difference in the number of CD3+/mm^2^ T lymphocytes in CT between LVT patients and control group was statistically significant (*p <* 0.001, 95% confidence interval) (Table 2 and Figure 4B).

### 4.4. Treatment and Follow-up

All LVT patients received conventional medical treatment for arrhythmias [13] in addition to immunosuppressive therapy including 1 mg/kg prednisone daily for four weeks followed by 0.33 mg/kg daily for five months and 2 mg/kg azathioprine daily for six months in case of virus-negative myocarditis according to the TIMIC trial (Tailored Immosuppression in Inflammatory Cardiomyopathy) [14] and oseltamivir in case of influenza A myocarditis [15]. Patients were followed for six months and the resolution of ventricular tachyarrhythmias on Holter monitoring was classified as response to therapy.

The eight patients with immune-mediated myocarditis were treated with immunosuppression [14]. Immunosuppression was successful in seven, with disappearance of ventricular arrhythmias on ECG as well as on Holter monitoring at six-month follow-up. None of the patients on immunosuppression had major drug-related side effects requiring therapy withdrawal.

The two patients with influenza A virus-related myocarditis were successfully treated with oseltamivir. The other two patients with virus-positive myocarditis died because of ventricular fibrillation. Interestingly, in the patient with HSV2-related myocarditis, the viral infection caused a fulminant myocarditis with infiltration of cardiac conduction tissue and ganglia inducing electrical instability and sudden death.

Overall, molecular characterization of CT damage in our patient population was followed by resolution of ventricular arrhythmias in 75% of cases.

## 5. Discussion

The arrhythmic phenotype of myocarditis is difficult to diagnose as the prodromal phase of infection is often subclinical or absent, cardiac anatomy and function are usually preserved and CMR sensitivity is as low as 40% [3]. In this instance, clinical approach is usually conservative and after a normal coronary angiography, a symptomatic treatment is adopted including an antiarrhythmic therapy and/or an ICD implantation. Indeed, both solutions present some limitations as the former can be ineffective exposing the patient to the risk of cardiac arrest while the latter can be excessive since a spontaneous resolution of inflammation occurs in as many as 50% of cases, leaving the patient with a permanent unnecessary device. Finally, cardiac dilatation and dysfunction can follow the initial electrical instability. All these considerations lead to the need for a morpho-molecular characterization of myocardial substrate and to indications for a more specific therapy. To this end, endomyocardial biopsy is recommended by major scientific statements [1,16,17] as the feared complications from this tool are very low (<1%) and transient [18] even when extended to the left ventricle [6]. In addition, ventricular endomyocardial biopsy, particularly when applied to ventricular septum, may allow the observation of peripheral sections of the His-Purkinje network [4] that is hypothesized to be affected in the arrhythmic phenotype of myocarditis.

Indeed, while the limited compromise of contractility in this form is believed to be due to focal inflammation of the myocardium, the prominent electrical instability would be the consequence of an elective involvement of conduction tissue (CT). Up to now, only occasional single case reports [19,20] document CT involvement in this type of myocarditis leaving uncertainties in cause and outcome.

The present report presents the morpho-molecular study of CT included in 12 out of 42 consecutive myocarditis patients (28.5%) presenting with an arrhythmic phenotype including ventricular tachycardia or fibrillation. In addition to myocardial real time PCR for the major cardiotropic viral genomes, CT was characterized by type of inflammatory cells, presence of viral proteins and expression of TLR4 that contribute to the exposition of new immunogenic antigens. In our study, molecular CT characterization together with myocardial PCR helped to define a specific anti-viral or immunosuppressive regimen in addition to the anti-arrhythmic therapy. Specifically, oseltamivir was revealed to be effective in preventing arrhythmic recurrences in patients with CT inflammation by influenza virus while immunosuppressive therapy was usually effective in those subjects where inflamed CT was virus negative and exhibited overexpression of TLR4. Unfortunately, patients with CT infection by adenovirus and HSV2 died. Overall, 75% of patients with life-threatening ventricular arrhythmias due to myocarditis and treated on the basis of morpho-molecular characterization of inflamed CT could completely recover with no arrhythmias registered at a six-month follow-up.

Finally, in the control group with myocarditis and no repetitive ventricular tachyarrhythmias, CT appeared to be unaffected by inflammatory infiltration and damage (*p <* 0.001).

In conclusion, arrhythmic phenotype of myocarditis is associated with elective inflammation/infection of CT. Molecular characterization of CT damage may lead to pharmacologic control of arrhythmias in 75% of cases. A prospective randomized trial including a larger number of patients and an alternative (i.e., immunosuppressive vs antiarrhythmic) therapy, is necessary to confirm the present results.

## Figures and Tables

**Figure 1 jcm-09-03470-f001:**
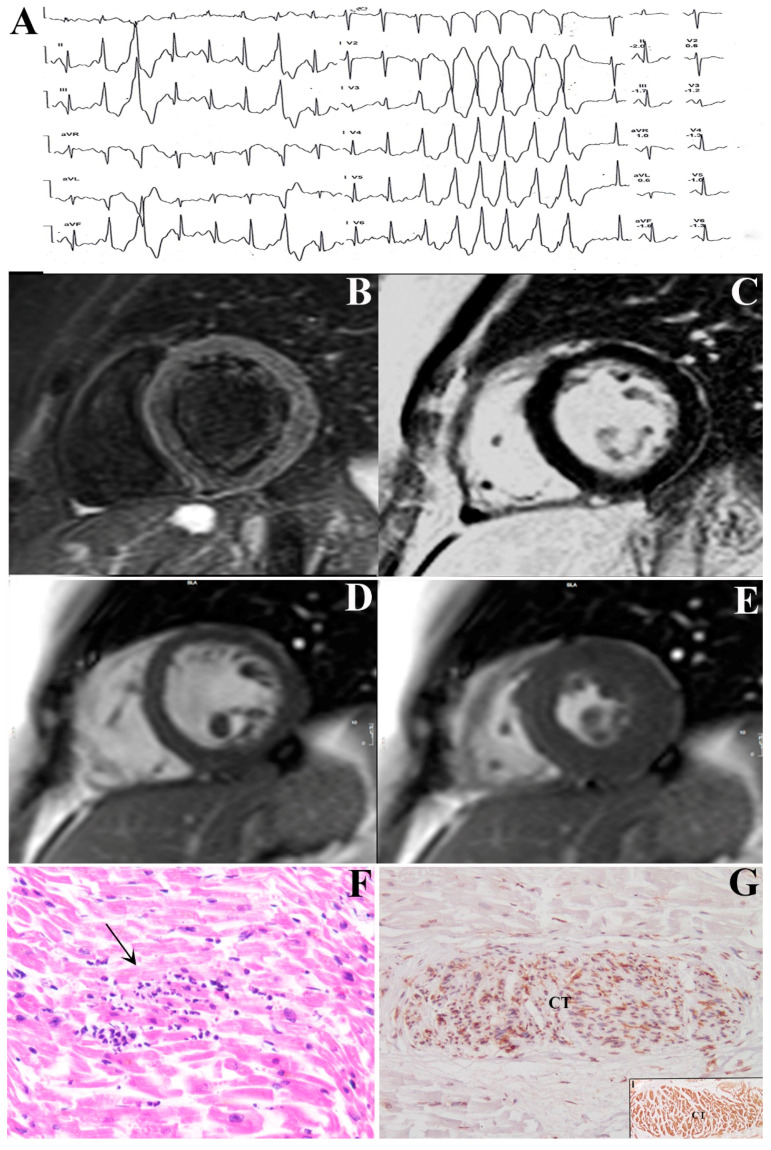
44-year old woman (pt 4 of Table 1) presenting with non-sustained ventricular tachycardia panel (**A**), normal cardiac magnetic resonance (CMR) cardiac parameters on short-axis cineMR images acquired on diastolic panel (**D**) and systolic panel (**E**) phases and no myocardial edema on Short Tau Inversion Recovery (STIR) T2 weighted images panel (**B**) nor late gadolinium enhancement (LGE) signal abnormalities panel (**C**). The patient at left ventricular endomyocardial biopsy shows focal lymphocytic myocarditis panel (**F**) and extensive lymphocytic T (CD45RO+) infiltration of conduction tissue panel (**G**) identified by HCN4 (Hyperpolarization-activated and cyclic nucleotide-gated) immunostaining (panel (**G**) insert i).

**Figure 2 jcm-09-03470-f002:**
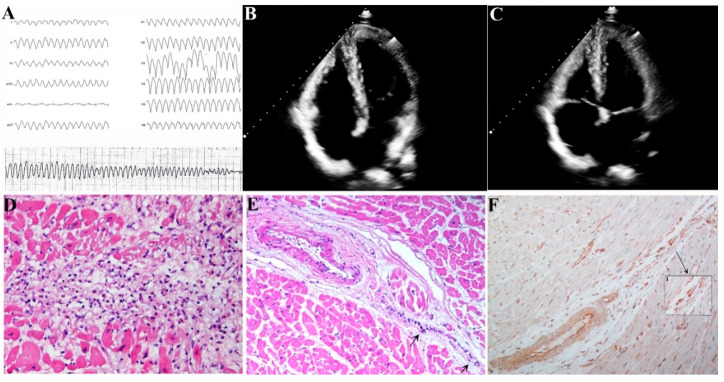
41-year old woman (pt 3 of Table 1) with sustained ventricular tachycardia degenerating into ventricular fibrillation panel (**A**), presenting normal echocardiographic parameters panels (**B**) and (**C**). At autopsy an active lymphocytic myocarditis panel (**D**) was associated with clear inflammatory infiltration of conduction tissue (CT) (panel (**E**), arrow). Myocardial PCR was positive for HSV2 infection and the immune-histochemistry against HSV2 glycoprotein B panel (**F**) showed an overstaining of CT cell nuclei (insert i).

**Figure 3 jcm-09-03470-f003:**
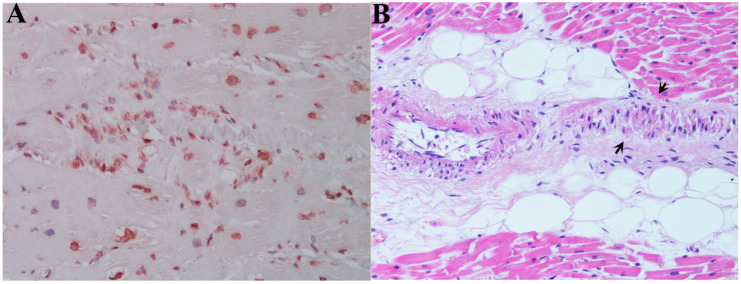
(**A**) Left ventricular endomyocardial biopsy of control group shows focal lymphocytic myocarditis (CD45RO+) (200× magnification) and absence of lymphocyte infiltration of conduction tissue panel (**B**) (EE, 200× magnification).

**Figure 4 jcm-09-03470-f004:**
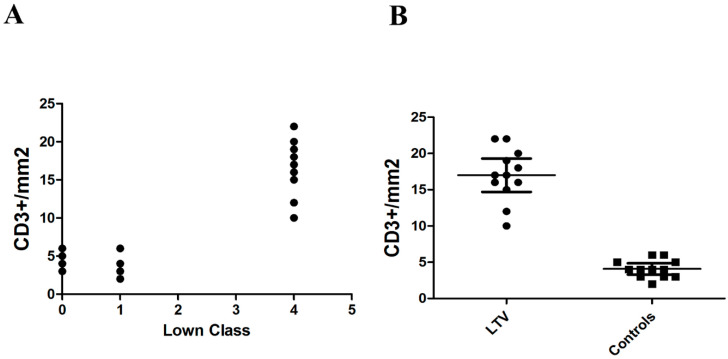
(**A**) Positive correlation between number of CD3+/mm^2^ cells in CT and severity of Lown Class. (**B**) Statistically significant difference in number of CD3+/mm^2^ cells between LVT and control group.

**Table 1 jcm-09-03470-t001:** Baseline clinical parameters and outcome of patient population.

	Pt 1	Pt 2	Pt 3	Pt 4	Pt 5	Pt 6	Pt 7	Pt 8	Pt 9	Pt 10	Pt 11	Pt 12
Age/sex	66 M	21 F	41 F	44 F	52 F	24 M	46 M	38 M	71 F	42 M	52 M	56 M
Etiology	V (influenza A virus)	I	V (HHV2)	I	I	I	I	I	V (AV)	I	V (influenza A virus)	I
Flu-like syndrome	Yes	No	Yes	Yes	No	No	Yes	Yes	Yes	No	Yes	Yes
Clinical arrhythmic manifestation	sVT	VF	VF	nsVT	sVT	nsVT	sVT	nsVT	sVT	sVT	sVT	nsVT
LVEF, % 2D-echo	53	50	50	52	60	60	60	60	55	68	62	64
LVEDD, mm 2D-echo	54	48	50	56	51	41	54	52	51	47	53	49
MWT, mm 2D-echo	11	8	8	8	9	17	11	12	10	9	10	9
LVEDV, mm/m^2^ 2D-echo	65	68	62	99	68	56	82	68	67	61	64	69
LVESV, mm/m^2^ 2D-echo	36	34	31	47	27	20	33	16	30	19	25	25
Edema cMRI	(−)	(−)	N/P	(−)	(−)	(−)	(−)	(−)	(−)	(−)	(−)	(-)
EGE, cMRI	(−)	(−)	N/P	(−)	(−)	(−)	(−)	(−)	(−)	(−)	(−)	(-)
LGE, cMRI	(+)	(+)	N/P	(−)	(+)	(+)	(−)	(−)	(−)	(−)	(+)	(+)
Therapy	Oseltamivir	P + A	-	P + A	P + A	P + A	P + A	P + A	-	P + A	Oseltamivir	P + A

Legend: V—viral; I—idiopathic; HHV2—human herpesvirus 2; AV—adenovirus; sVT—sustained ventricular tachycardia; nsVT—non-sustained ventricular tachycardia; VF—ventricular fibrillation; LVEF—left ventricular ejection fraction; LVEDD—left ventricular end-diastolic diameter; MWT—maximal wall thickness; cMRI—cardiac magnetic resonance imaging; LVEDV—left ventricular end-diastolic volume; LVESV—left ventricular end-systolic volume; EGE—early gadolinium enhancement (suggestive of hyperemia); LGE—late gadolinium enhancement (suggestive of fibrosis/necrosis); P + A—prednisone plus azathioprine; N/P—not performed.

**Table 2 jcm-09-03470-t002:** Comparison between LVT and control group.

	LVT Group	Control Group	*p* Value
Age	46.1 ± 14.8	44.3 ± 12.7	0.7507
Sex	7 M, 5 F	9 M, 3 F	0.7104
LVEF *, % 2D-echo	57.8 ± 5.8	35.4 ± 10.8	*p <* 0.0001
LVEDD ^†^, mm 2D-echo	50.7 ± 4.1	58.3 ± 3.8	*p <* 0.0001
MWT ^‡^, mm 2D-echo	10.1 ± 2.4	10.1 ± 1.8	0.7686
LVEDV ^§^, mm/m^2^ 2D-echo	69.7 ± 10.5	112.5 ± 13.2	*p <* 0.0001
LVESV ^#^, mm/m^2^ 2D-echo	29.7 ± 10.8	72.9 ± 16.2	*p <* 0.0001
CD3+ lymphocytes, mm^2^	17.0 ± 3.6	4.0 ± 1.2	*p <* 0.0001

Legend: * left ventricular ejection fraction; ^†^ left ventricular end-diastolic diameter; ^‡^ maximal wall thickness; ^§^ left ventricular end-diastolic volume; ^#^ left ventricular end-systolic volume. *p <* 0.05 is considered statistically significant.

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
