# Peer review of "Inflammation of Conduction Tissue in Patients with Arrhythmic Phenotype of Myocarditis"

_jcm, 2020, doi:10.3390/jcm9113470_

Round 1

Reviewer 1 Report

In this work, A Frustraci and col studied the cardiac biopsies of 12 patients with myocarditis and cardiac arrhythmias. They found that 12/12 biopsies had signs of inflammation in the conduction tissues and conclude that "arrhythmic phenotype of myocarditis is caused by elective inflammation/infection of CT. Molecular characterization of CT damage may lead to pharmacologic control of arrhythmias in 75% of cases"

The topic is original but in this current state, results are unsufficient to support the conclusions.

I have two major concerns:

Stating that arrhythmic phenotype is caused by inflammation in the CTs would require to demonstrate at least an association or a correlation between them. Therefore authors must demonstrate that in a control group with myocarditis but without arrhythmia, there is no or less inflammation in the CT.

Demonstrating causality would then require to document that suppressing the cause would suppres the consequence. in the second part of the conclusion authors state that "Molecular characterization of CT damage may lead to pharmacologic control of arrhythmias". To demonstrate it, they would have to perform a trial that would compare a tailored pharmacologic strategy based on the molecular characterization of the CT with a conventionnal standard of care. Therefore it seems more appropriate to tell that patients have been managed using a molecular characterization of the cardiac biopsies and that clinical improvement was documented in 75% of them. Furthermore, the molecular characterization was performed on cardiac biopsies that were not restricted to the CTs. Conclusions have to be smoothed on this point as well.

Author Response

We thank the reviewer for His/Her comments.

A control group of 12 pts with myocarditis , no repetitive ventricular arrhythmias (Lown class ≤3)and inclusive of uninflamed CT is considered in the abstract and test. CD3+ lymphocytes /2mmin in CT were counted and a statistically significant difference (p< 0.001) between those with and without arrhythmic presentation is demonstrated.

To support the difference an additional figure (figure 3) is added showing in the control groups with myocarditis and no repetitive ventricular arrhythmias a preserved, unflammed CT.

Reviewer 2 Report

In this manuscript the authors present their findings in regards to to conduction tissue inflammation in endomyocardial biopsies from patients presenting with myocarditis and who had an arrhythmic phenotype. Evidence of inflammation were described in all 12 patients with arrhythmic presentation of myocarditis that were biopsied. Eight patients had negative PCR for viral genomes and in the remaining, Influenza A virus (n=2), HSV2 (n=1), and adenovirus (n=1) genomes were found.

Understanding the pathological substrate in patients presenting with myocarditis and arrhythmias is key as ventricular arrhythmias significantly contribute to patient mortality.

This study provides some useful insights towards better understanding this issue but I do have a major concern regarding the methodology and the conclusions that can be derived from this study. The authors conclude that "arrhythmic phenotype of myocarditis is caused by elective inflammation/infection of CT" and "Molecular characterization of CT damage may lead to pharmacologic control of arrhythmias in 75% of cases". However there are no correlation or associations studied that can support these conclusions.

  • In order for a study to demonstrate that the arrhythmic phenotype of myocarditis is linked to inflammation of the conduction tissue, cases without the arrhythmic phenotype should also have been studied and compared.
  • Furthermore, in order to suggest that it was the treatment that lead to pharmacologic control of the arrhythmias, a non-treated branch would be required as well.

Essentially this is a case series of patients with arrhythmic phenotype of myocarditis that provides some useful insights worth sharing with the international community but is not able to make associations and correlations. I recommend that the authors change the manuscript and their conclusions accordingly.

Author Response

(The authors gave the same response as above.)

Round 2

Reviewer 1 Report

In this revised manuscript, the authors added a control group of 12 patients with myocarditis and no arrhythmic events. These supplemental analyses improve the manuscript and strengthen the author conclusions.

I still have minor suggestions:

  • in the abstract, I suggest to rephrase and clarify the sentence "Control was uninflamed CT from 12 pts with myocarditis and no LVT" into "In 12 patients with myocarditis and no LVT, CT were uninflammed (P...)". These analyses should be presented in the Results section of the abstract and the manuscript.
  • conclusion sentence of the abstract has been smoothed regarding the causality between inflammation and arrhythmia. Similar modification should be performed accordingly in the conclusion of the manuscript (p4)
  • in the discussion, authors state that specific anti-viral / immunosuppressive therapy would treat effectively these patients. Once again, such conclusions could not be drawn from the presented observational data and would require a specific trial. Discussion and conclusions should be smoothed accordingly.

Author Response

In this revised manuscript, the authors added a control group of 12 patients with myocarditis and no arrhythmic events. These supplemental analyses improve the manuscript and strengthen the author conclusions.

I still have minor suggestions:

  • in the abstract, I suggest to rephrase and clarify the sentence "Control was uninflamed CT from 12 pts with myocarditis and no LVT" into "In 12 patients with myocarditis and no LVT, CT were uninflammed (P...)". These analyses should be presented in the Results section of the abstract and the manuscript.
  • Reply: We thank the reviewer for the suggestion. The sentence has been modified accordingly in the abstract and text.

  • conclusion sentence of the abstract has been smoothed regarding the causality between inflammation and arrhythmia. Similar modification should be performed accordingly in the conclusion of the manuscript (p4)
  • Reply: The conclusions are attenuated requiring a prospective randomized trial with a large number and an alternative therapeutic option (immunosuppression vs antiarrhythmic therapy)
  • in the discussion, authors state that specific anti-viral / immunosuppressive therapy would treat effectively these patients. Once again, such conclusions could not be drawn from the presented observational data and would require a specific trial. Discussion and conclusions should be smoothed accordingly.

Reply. These considerations are reported in the conclusions.

Reviewer 2 Report

The authors seem to have changed the methodology into a case-control study to further support their conclusions. However, I have a few concerns:

1) Where was the control group derived from? Is it from the total 420 myocarditis patient cohort? How many patients from that cohort had CT sections in the absence of arrhythmias in total?

2) What were the criteria for selection of the control group cases? Were they age- and sex- matched?

3) A table that compares the demographics and clinical information between the control group and the LTV group should be presented with appropriate statistical comparisons in order to identify potential bias in the comparison.

4) In their reply, the authors state: "CD3+ lymphocytes /2mmin in CT were counted and a statistically significant difference (p< 0.001) between those with and without arrhythmic presentation is demonstrated.". I could not find this in the manuscript. The actual numbers between the control group and the LTV group should be presented in a graph with individual data points, mean value and confidence intervals as this is the key point of this paper.

Author Response

The authors seem to have changed the methodology into a case-control study to further support their conclusions. However, I have a few concerns:

  • Where was the control group derived from? Is it from the total 420 myocarditis patient cohort? How many patients from that cohort had CT sections in the absence of arrhythmias in total?

Reply:  The original pt population was that of 420 with myocarditis. Eightyfour patients presented at histology the inclusion of  peripheral sections of CT. Twelve patients had LVT and inflammed CT. Seventytwo pts had a cardiomyopathic or infarct-like myocarditis, no LVT and Lown class between 0 and 3. Twelve consecutive patients  matched for sex and age and with CT included in myocardial tissue were considered as control group.

  • What were the criteria for selection of the control group cases? Were they age- and sex- matched?

Reply: pts were matched for sex, age, and no repetitive ventricular arrhythmias at Holter recording.

  • A table that compares the demographics and clinical information between the control group and the LTV group should be presented with appropriate statistical comparisons in order to identify potential bias in the comparison.

Reply: A table comparing LVT patients and controls has been included (Table 2).

  • In their reply, the authors state: "CD3+ lymphocytes /2mmin in CT were counted and a statistically significant difference (p< 0.001) between those with and without arrhythmic presentation is demonstrated.". I could not find this in the manuscript.

Reply: The sentence is reported in the text page 3 line 53.

  • The actual numbers between the control group and the LTV group should be presented in a graph with individual data points, mean value and confidence intervals as this is the key point of this paper.

Reply: A graphic with quantification of CD3+ cells between study population and control are now provided.(see Figure 4).